## Classics

stem cells; root apical meristem; laser ablation; positional information; auxin.

**Corresponding author:**
Ikram Blilou;
Email: ikram.blilou@kaust.edu.sa

# A blast from the past: Understanding stem cell specification in plant roots using laser ablation

Wouter Smet and Ikram Blilou

Biological and Environmental Science and Engineering (BESE) Division, Plant Cell and Developmental Biology, King Abdullah University of Science and Technology (KAUST), Thuwal, Kingdom of Saudi Arabia

## Abstract

In the Arabidopsis root, growth is sustained by the meristem. Signalling from organiser cells, also termed the quiescent centre (QC), is essential for the maintenance and replenishment of the stem cells. Here, we highlight three publications from the founder of the concept of the stem cell niche in Arabidopsis and a pioneer in unravelling regulatory modules governing stem cell specification and maintenance, as well as tissue patterning in the root meristem: Ben Scheres. His research has tremendously impacted the plant field. We have selected three publications from the Scheres legacy, which can be considered a breakthrough in the field of plant developmental biology. van den Berg et al. (1995) and van den Berg et al. (1997) uncovered that positional information-directed patterning. Sabatini et al. (1999), discovered that auxin maxima determine tissue patterning and polarity. We describe how simple but elegant experimental designs have provided the foundation of our current understanding of the functioning of the root meristem.

## 1. Introduction

Because plant cells are immobilised in their tissue context due to their rigid cell walls, the formation and growth of the plant body require a precise temporal and spatial coordination of cell divisions and cell expansion. Post embryonically, continuous growth and organogenesis are driven by apical meristems in the root and shoot. The Arabidopsis root apical meristem (RAM) has been a model for plant patterning since the early 1990s, with one of the main attractive features being its anatomical simplicity (Dolan et al., 1993; Scheres et al., 1994) (Figure 1a). Because of its highly stereotyped organisation, cells in the Arabidopsis root can be easily recognised and traced back to the stem cells of origin. In the RAM, central stem cells surround an organising centre known as the quiescent centre (QC). Through asymmetric cell division, these stem cells provide progenitors for the individual cell types within the root (Figure 1a). Following successive rounds of divisions, cells exit the meristem and differentiate further. The balance between cell division and differentiation defines zonation within the root (Figure 1b) (Blilou et al., 2005; Dello Ioio et al., 2008, 2007; Galinha et al., 2007; Grieneisen et al., 2007; Moubayidin et al., 2013). Because of its structural simplicity and transparency, the Arabidopsis root has been an ideal system to address fundamental questions on how stem cells are specified and maintained during growth and regeneration, which is key to developmental plasticity.

Here, we summarise and discuss three publications that have laid the foundation that helped the root biology community understand the role of the QC as an organiser centre and its function in maintaining the stem cells in the root meristem. These publications are, in our opinion, 'classics' in plant developmental biology. The first publication by van den Berg et al. (1995) focuses on cell fate acquisition in the root meristem and whether this is determined by clonal origin or positional control. The second publication by van den Berg et al. (1997) reveals insight into the QC's function and how it maintains stem cell activity in the root stem cell niche. Third, we discuss the publication of Sabatini et al. (1999), highlighting the importance of auxin maxima in defining and maintaining the root stem cell niche. We will outline the regulatory modules and pathways identified as signals that control cell division and differentiation within the root stem cell niche.

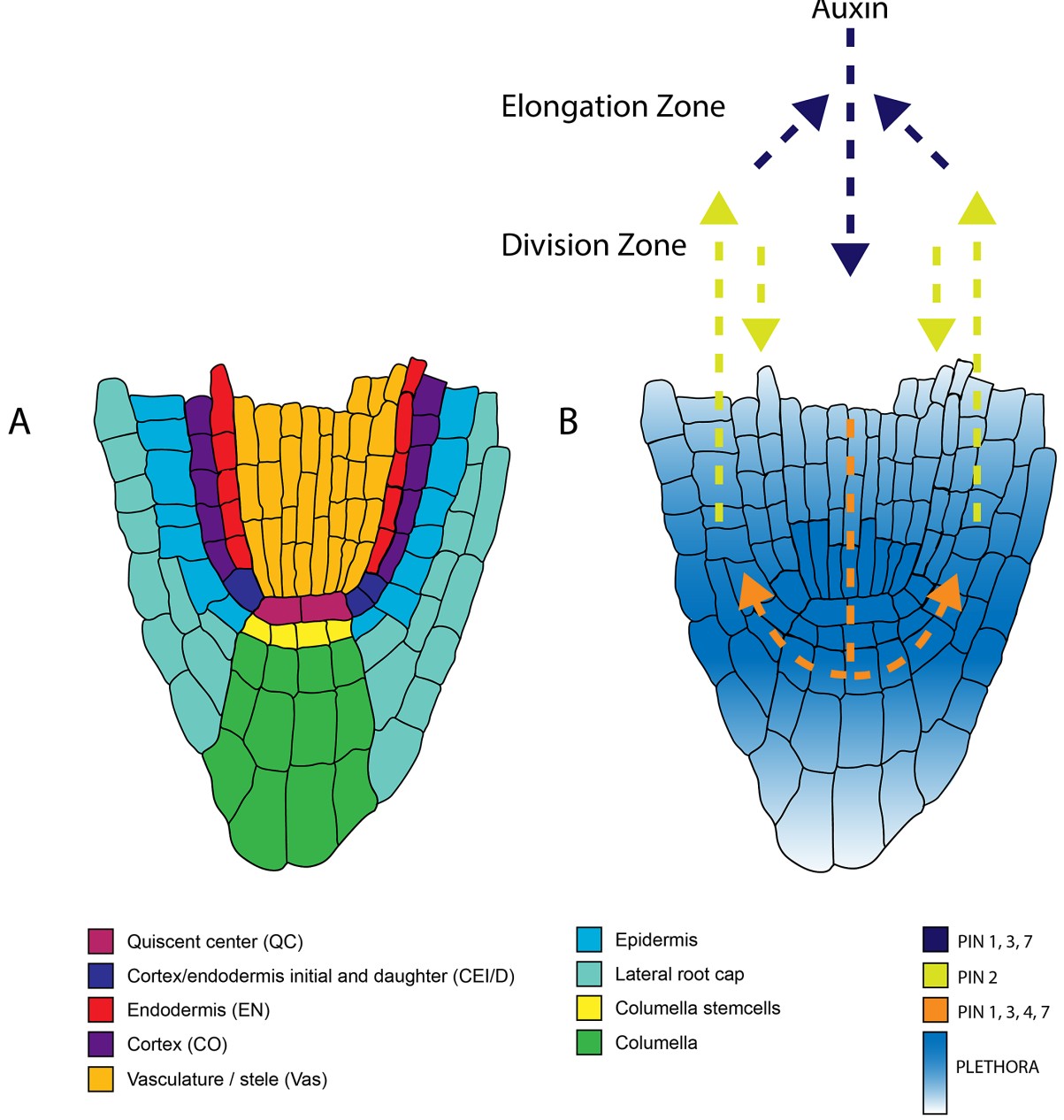

**Figure 1.** Anatomy of the root apical meristem and pathways involved in patterning and specification of the root apical meristem. A. Detailed overview of the different cell types within the root apical meristem. B. Auxin transport and the underlying PLETHORA gradient. Arrows indicate the direction of the auxin flux directed by the PINproteins corresponding to the colors in the legend. The PLETHORA gradient is represented by the white-blue gradient. Strongest expression is found in the QC and surrounding stem cells and decreases further away from the stem cells.

## 2. Cell fate in the Arabidopsis root meristem is determined by directional signalling (van den Berg et al., 1995)

### 2.1. Defining the Arabidopsis fate map

In animal systems, evidence that a subset of cells controls the fate of the neighbouring cells through inducing signals was established in the 1930s (Jacobson & Rutishauser, 1986; Jessell & Melton, 1992; Spemann, 1938). These signals could diffuse between cells and establish different concentrations within a tissue. Differences in signal concentration can induce different cell fates within the tissue (Jessell & Melton, 1992). In plants, there had been indirect evidence where surgical experiments in the shoot apical meristem suggested that shoot apical meristem cells do not have a predictable destiny

and favoured the hypothesis that position is a determinant for the acquisition of cell fate (Jegla & Sussex, 1989; McDaniel & Poethig, 1988; Pilkington, 1929).

During development, the function of genes in determining distinct tissue types can be assessed by studying their respective mutants (Ingham, 1988; Mayer et al., 1991). However, such analyses can be conducted only if a basic description of how cells and tissues are precisely organised within an organ is available. In the Arabidopsis root meristem, information obtained from histological data, clonal analysis, and electron microscopy have defined the cellular organisation of Arabidopsis root meristem and constructed its fate map (Dolan et al., 1993; Scheres et al., 1994). All tissue types can already be identified in the heart-stage embryos. Each cell file

can be traced to initial/stem cells, which divide to generate a new stem cell and a daughter cell that undergoes successive divisions and differentiates into a cell with a different fate. The simple and highly organised pattern of Arabidopsis root meristem was key to accelerating genetic research and enabled sophisticated experimental designs to be applied in root biology. The highly organised structural pattern allowed the implementation of laser ablation experiments of single cells in the Arabidopsis root meristem, which has provided the first evidence that shed light on the importance of cell–cell signalling for cell differentiation and cell fate acquisition in Arabidopsis (van den Berg et al., 1995). Using a simple but elegant experimental design, it was demonstrated that the position of a cell is important for cell fate determination in the root meristem.

### 2.2. Directional signalling defines cell fate in the root meristem

The work by van den Berg et al. (1995) determined that positional control is important for cell fate determination and that directional signalling guides tissue patterning within the root meristem. These conclusions were obtained based on a simple experimental design where a laser is used to kill/damage a selected cell or group of cells, after which the behaviour of the surrounding cells is monitored (Figure 2a). Upon laser ablation of cells within the RAM, dead cells are compressed, and their position is filled by neighbouring cells. Determining the new fate of the invading cell and which cell file divides to fill this position helped distinguish between the clonal origin or position hypotheses. First, the organising centre cells, or QC cells, were ablated. The dead QC cells were displaced towards the root tip. After ablation, they observed that the proximal vascular cells take up the position of the former QC. These cells lose their vascular identity as they cease to express the vascular marker and switch to root cap fate. This seemingly simple experiment led to two significant findings. First, the clonal boundary set by the first zygotic division, separating future vascular and root cap cells, does not restrict developmental potential. Additionally, they concluded that information guiding cell fate along the apical-basal axis in the root tip must be permanently present.

Next, the authors ablated initial cells generating the different cell lineages within the RAM to assess whether the positional information also determined cell fate in the radial axis. This revealed that the cell inwards to the ablated cell divides, takes up its position, and switches the cell fate according to the newly occupied position (Figure 2a,b). More recent work by Marhava et al. (2019) has shown that this preference for the inner cell to replace the open position is true for cells throughout the root meristem. Several studies in the last decades have further elucidated the molecular players involved in this regeneration process (Efroni et al., 2016; Hoermayer et al., 2020; Marhava et al., 2019; Moreno-Risueno et al., 2015; Sabatini et al., 1999; Sena et al., 2009; Xu et al., 2006; Zhou et al., 2019). If initial cells behave according to their position, the authors reasoned that there must be signalling that ensures that growth and patterning are maintained correctly. To test the directionality of signalling, they ablated the daughter cell of cortical initials before their asymmetric periclinal division that usually gives rise to the cortex and endodermis. After ablation, they monitored the behaviour of the underlying cortical initial cells (Figure 2c). Interestingly, the underlying initial cell still underwent its regular sequence of divisions. They reasoned that because the cell is connected via a three-way junction, it can still receive the correct positional input from undergoing periclinal cell division. To address this, all neighbouring three cortex initial daughter cells

were ablated instead (Figure 2d). The cortex initially divides but fails to undergo the periclinal cell division to generate cortex and endodermis. From this, the authors concluded that information guiding the allocation of cell fate in the radial plane is propagated through an individual cell layer and is directed towards the tip. Together, the data in this paper suggest that positional information can be transferred from mature cells to stem cells to control tissue patterning in the root meristem.

## 3. Short-range control of cell differentiation in the Arabidopsis root meristem (van den Berg et al., 1997)

### 3.1. QC cells inhibit differentiation of contacting columella initials

van den Berg et al. (1997) wanted to get insights into how the balance between division and differentiation is maintained within the root meristem. Each initial cell within the root apical meristem is a stem cell for its respective cell lineage. The initials are in contact with cells in the centre of the root stem cell niche, also known as the quiescent centre. At the time, it was proposed that these cells act either as a reservoir of stem cells or as an organising centre. Their previous publication (van den Berg et al., 1995) showed that upon complete ablation of the QC, it is rapidly replaced by cells from the stele. This indicated that the QC is not the sole reservoir of stem cells; instead, surrounding cells can switch fate to fulfil this function upon QC damage. To study the function of the QC, the authors again resorted to ablation, but in this case, they only ablated one of the two QC cells, resulting in a slower replacement of the QC cells (Figure 2e).

Upon ablation of single QC cells, the contacting columella cells stopped dividing and differentiated, which was marked by the appearance of starch granules (Figure 2f). This only occurred in the cells directly in contact with the ablated cells. This was the first evidence indicating that the connection of the QC is required for the columella initial cells to remain in their initial state. From these observations, the authors generated three models for QC functioning:

- The QC separately promotes cell division and inhibits differentiation.
- The QC promotes cell divisions, which in turn inhibits differentiation.
- The QC inhibits differentiation, which in turn promotes cell division.

To distinguish between the proposed models, the authors first monitored the QC activity in mutants lacking post-embryonic cell division. Here, the QC ablation also resulted in the differentiation of the columella initial cells just as in wild-type conditions, indicating that it is not the cell division potential that is required for maintaining the initial state and, instead, it is most likely the signalling from the QC that is needed for preventing differentiation.

### 3.2. The QC controls the differentiation state of multiple initials

To see if the inhibition of differentiation by the QC affects other initial cells, the authors explored the effect of QC ablation on the cortex initial cells. These usually first undergo an anticlinal division, after which the daughter cells undergo an asymmetric periclinal division that generates the endodermis and the cortex. Upon ablation of the QC, the contacting cortical initial cells fail to divide anticlinally and, instead, immediately undergo periclinal division

Van den Berg  et al. 1995.

Van den Berg  et al. 1997.

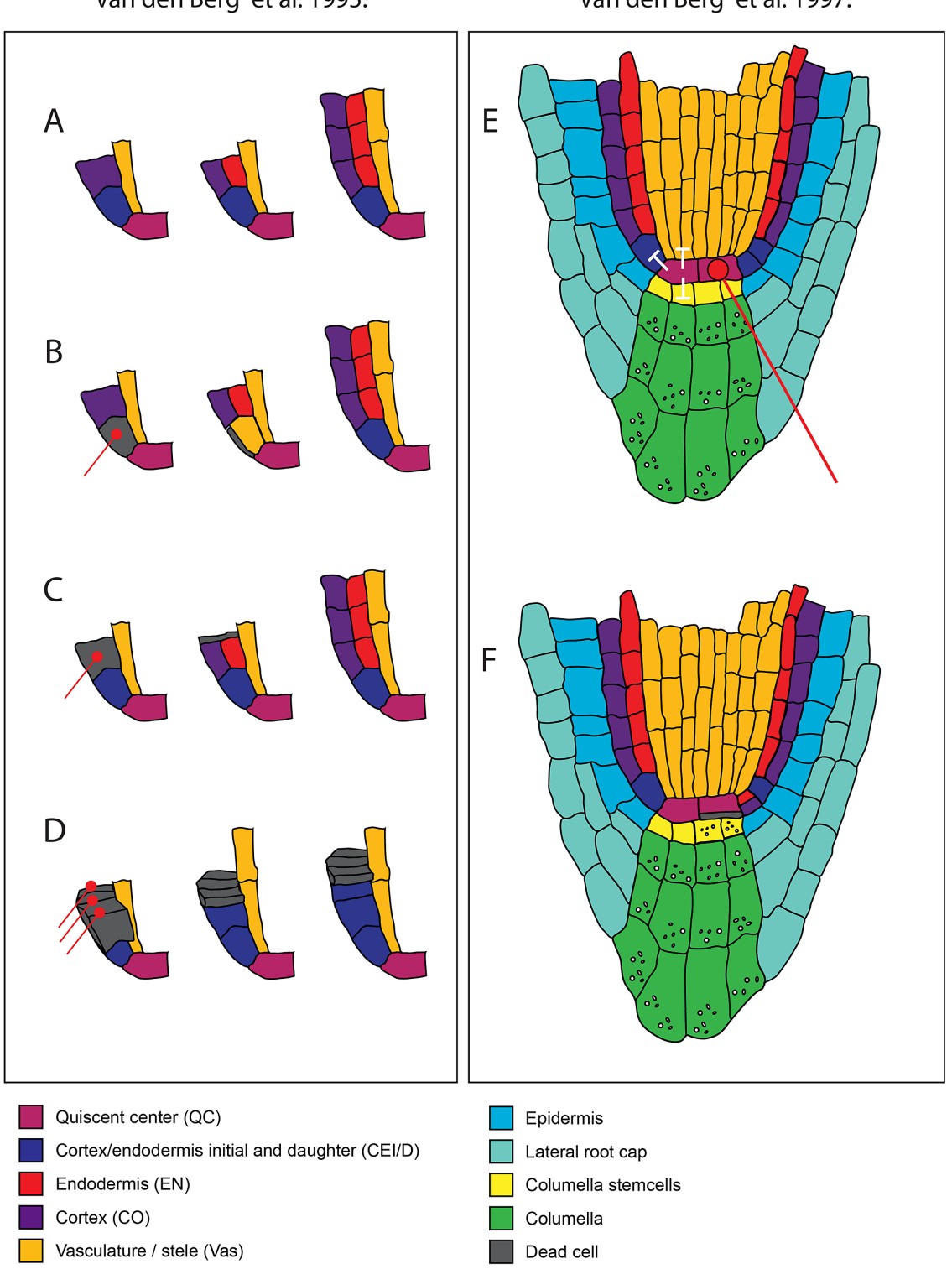

**Figure 2.** Model describing the findings of van den Berg et al. 1995 and 1997 showing the effect of laser ablation on patterning of the surrounding cells in the stem cell niche. A. Sequence of divisions from the cortex endodermal initial that generate the cortex and endodermis in normal conditions. B. Ablation of the cortex endodermal initial cell leads to its cell death and the cell will be compressed. It will be replaced by a cell from the vasculature but this does not affect the sequence of division of the overlaying cells. C. Ablation of the cortex endodermal daughter cell leads to its cell death and the cell will be compressed. It will be replaced by a cell from the vasculature but this does not affect the sequence of division of underlying cells. D. Ablation of three cortex endodermal daughter cells overlaying the initial cell. The underlying initial cell now undergoes anticlinal division but fails to undergo the subsequent periclinal division that normally results in the formation of both cortex and endodermis. E. Root apical meristem upon ablation of one QC cell. White arrows indicate the non-cell autonomous inhibition of differentiation by the QC on the surrounding stem cells. F. After laser ablation of one QC cell the contacting columella stem cells differentiate, marked by the formation of starch granules. Also, the contacting cortex endodermal initial undergoes periclinal division.

and differentiate into endodermis and cortex. In contrast, cortical initial cells contacting intact QC cells undergo normal division. These results show that the QC controls the differentiation status of multiple contacting initial cells (Figure 2f). Further analysis after ablation in combination with using 3H-thymidine revealed that cells surrounding the QC showed similar rates of cell division compared to non-ablated cells. Only the columella appeared to be affected in its cell division rate.

Altogether these data suggest that the QC controls the initial states of the surrounding cell not by regulating cell division in the columella but mainly by inhibiting differentiation. Extensive studies have followed up based on the outcome of these two publications, and the identification of genes involved in stem cell maintenance and tissue patterning, pathways regulating root meristem development have been identified. Those include transcription factors, receptor kinase signalling, longitudinal and radial gradient established by transcription factors, and plant hormones (Aida et al., 2002; Cruz-Ramírez et al., 2012; De Smet et al., 2008; Long et al., 2015; Mähönen et al., 2014; Sarkar et al., 2007; Stahl et al., 2009, 2013; Wildwater et al., 2005).

## 4. An auxin-dependent distal organiser of pattern and polarity in the Arabidopsis root (Sabatini et al., 1999)

### 4.1. A distal auxin maximum in the Arabidopsis root

The previous two publications highlighted the importance of positional signalling within the Arabidopsis RAM to instruct its maintenance and patterning. During development, tissue patterning and polarity are often achieved by the asymmetric distribution of molecules. How cells may acquire different fates in response to varying concentrations of an endogenous signal and how these signals are spatially and temporally restricted has been an extensive area of research (Aida et al., 2004; Blilou et al., 2005; Friml et al., 2003; Galinha et al., 2007; Gallagher et al., 2004; Gälweiler et al., 1998; Helariutta et al., 2000; Liu et al., 1993; reviewed in Yu et al., 2022). It is now clear that the proper distribution of such molecules in space and time is required to instruct cell fate decisions in responsive cells.

Pattern formation in plants is established during embryogenesis and is maintained in the meristems of both the root and shoot. How meristems organise and coordinate growth and differentiation in such a precise manner has been a long-standing question. Sabatini et al. (1999) provided a substantial gain in understanding the importance of having an auxin maximum within the stem cell niche for proper tissue patterning and polarity. Auxins have already been shown to be involved in numerous developmental processes, for example, cell division and elongation, and organ formation. This work was the first to use a collection of tissue-specific markers to understand the role of auxin distribution in tissue patterning. Several auxin-response transcription factors were described as required for root development (Berleth & Jurgens, 1993; Hardtke, 1998; Sessions et al., 1997; Ulmasov, Hagen, et al., 1997). Furthermore, auxin transport was shown to be necessary for patterning the Arabidopsis embryo, as interference with auxin transport resulted in embryonic patterning defects (Hadfi et al., 1998; Liu et al., 1993). Because of these correlations between auxin and patterning, the authors investigated whether the asymmetric distribution of auxin provides patterning information. Their model of choice was the root apical meristem because of its strict definitions of the cell lineages (Figure 3a) (Dolan et al., 1993; Scheres

et al., 1994). Studying auxin distribution required visualisation of the auxin itself; the *DR5::GUS* marker line was generated in a timely manner (T Ulmasov, Murfett, et al., 1997). This marker consists of seven tandem repeats of an auxin-responsive element combined with a minimal 35S CaMV promoter driving the expression of a $\beta$-glucuronidase reporter gene.

Expression analysis of *DR5::GUS* in the root revealed that high levels of auxin are present in the root tip compared to the mature zone (Figure 3b). The GUS maximum was located in the columella initial cells, while expression could also be detected in the QC and the differentiated columella root cap. Application of the polar auxin transport inhibitor Naphthylphthalamic acid (NPA) resulted in a shift and expansion of the *DR5* maxima, indicating that *DR5* activity depends on auxin transport (Figure 3c). On the other hand, the application of 2,4-dichloro phenoxy acetic acid (2,4-D) resulted in the staining of all cells. However, the external application of 2,4-D did not further elevate the expression in the QC and the columella. From this, the authors concluded that *DR5* marks the auxin levels in a cell-type independent manner.

### 4.2. Mutants in auxin response and transport have distal patterning defects

Auxin-response mutants had already been reported to affect patterning in plant development. Thus, the authors crossed and analysed the expression of *DR5::GUS* in the AUXIN RESPONSIVE FACTOR 5/MONOPTEROS mutant, *mpU21*, and AUXIN RESISTANT mutants *axr1/axr3* (Berleth & Jurgens, 1993; Leyser et al., 1993; Rouse et al., 1998). They found that all three mutants showed a decrease in *DR5* activity and correlated with defects in patterning or cell fate acquisition. These data indicated that the perception of an auxin peak in the root meristem is required for proper patterning and maintenance. At the time, studies of the polar auxin transporters were slowly emerging with influx transporters transporting auxin within the cells and the efflux carriers out of the cells (Gälweiler et al., 1998; Marchant, 1999; Müller et al., 1998). The authors evaluated the auxin distribution in both the influx carrier mutant *aux1* and the two efflux carrier mutants available at that time, *pin1* and *pin2*. While *aux1* did not show a mislocalisation of the auxin peak, most likely because of genetic redundancy, both *pin1* and *pin2* mutants displayed a mislocalisation of *DR5* distribution, with the *pin1-1* mutant having a distorted organisation of the columella and a mislocated *DR5* peak, and *DR5* being localised on one side of the lateral root cap and correlated with the agravitropic root phenotype in *pin2* mutants. With the mild phenotypes observed in the single mutants, the exact role of auxin transporters in tissue patterning was established only a few years later by creating higher-order mutants for both influx and efflux carriers (Blilou et al., 2005; Friml et al., 2003; Grieneisen et al., 2007; Ugartechea-Chirino et al., 2010).

### 4.3. Inhibition of polar auxin transport redirects distal pattern and polarity

To further investigate the correlation between the auxin maximum and patterning of the root tip, the authors examined whether high auxin levels were sufficient to direct cell fate specification and cell division orientation. Application of the auxin transport inhibitor NPA resulted in an expansion of the *DR5* expression domain into the flanking epidermis and, more proximal, cortex cells. This lateral

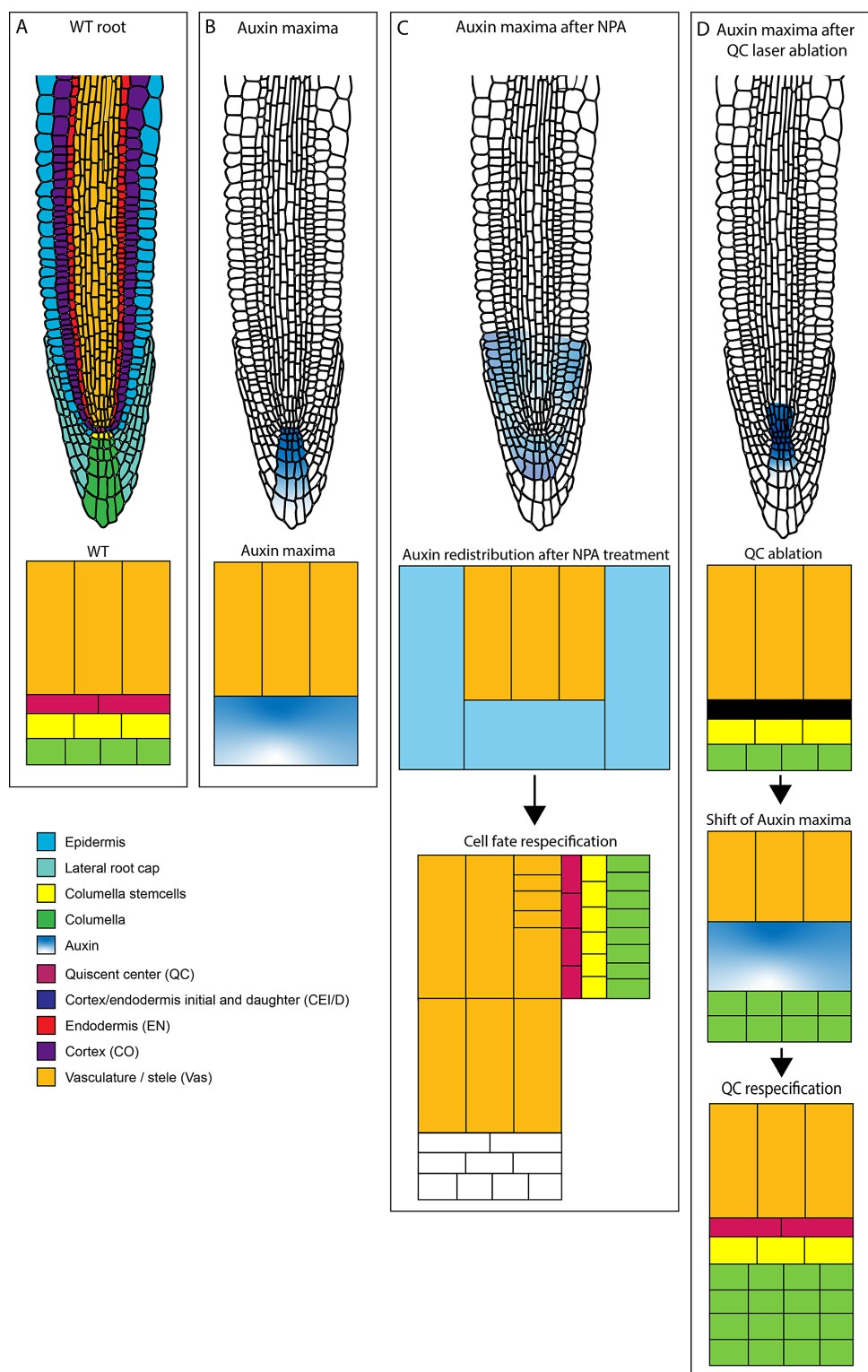

**Figure 3.** Model describing the findings in Sabatini et al. 1999. A. Scheme above showing the anatomy of a wild-type Arabidopsis root, Scheme below depicting a subset of cells of the QC, Vasculature, Columella stem cells and differentiated columella cells B. Scheme above showing the Auxin maxima found in the distal root tip using the DR5::GUS reporter, Scheme below showing auxin maxima in the same subset of cells as in A. C. Above scheme shows treatment with NPA expands the localization of the auxin maxima in the Arabidopsis root tip, below schemes showing the auxin redistribution and tissue repatterning. D. Above scheme showing the shifts of the auxin maxima in the distal root tip shifts upon laser ablation of the QC, Schemes below showing the correlation of the shift of the auxin maxima with the QC respecification and columella stem cell after ablation.

shift of *DR5* expression correlated with new divisions within the meristem and a shift in the division plane orientation (Figure 3c). The breakthrough of this study was that ectopic auxin accumulation and auxin redistribution within the root are sufficient to reorient cell division and induce distinct cell fate acquisition within the root meristem.

Interestingly, both distal patterning and polarity were redirected by the shift of the auxin maxima. This was the first evidence

that auxin orients distal patterning towards the vasculature and establishes an 'organiser' to induce distal tissues within the root meristem (Figure 3c).

The use of laser ablation methods to monitor changes in auxin distribution after QC ablation in addition to cell-type-specific markers like SCARECROW (SCR) and the endodermis/cortex identity enhancer trap line J0571 revealed that the accumulation of an auxin maxima relative to the vasculature predicts patterning and polarity (Figure 3d).

## 5. The stem cell niche: then and now

Before these three studies, the function of the QC within the root meristem was based on tissue anatomy and autoradiographic technologies, which had led to the concept that in most species, like the QC within the maize meristem has a slow division rate (reviewed in Steeves & Sussex (1989) based on Clowes (1958, 1956)). There have also been reports on the contribution of the QC in root recovery, and regeneration after damage caused by cold, x-irradiation, or root cap removal was already reported by Clowes (1976). It has also been proposed that a low division rate might be required to maintain the initial states of the stem cells, which would be analogous to stem cells in animals (Steeves & Sussex, 1989). The introduction of Arabidopsis as a model system for root development has revolutionised this field. The three studies presented here established that positional signalling controls root patterning by elegantly using laser ablation to study patterning and transcriptional reporters in the root apical meristem after wounding. The developmental programs that maintain the RAM are robust as the RAM is quickly repatterned after wounding. The recovery typically occurs within hours after reactivation of the stem cell transcriptional programs and accelerates the cell cycle's progression to assume the new cell fate according to the new position (Efroni et al., 2016; Marhava et al., 2019). This recovery process upon different types of wounding and the involved signalling pathways where auxin is a major player have been characterised in more detail in the last decades (Canher et al., 2020; Efroni et al., 2016; Heyman et al., 2013; Hoermayer et al., 2020; Liang et al., 2022, 2023; Marhava et al., 2019; Matosevich et al., 2020; Omary et al., 2023; Xu et al., 2006; Zhou et al., 2019).

Wounding induces a fast and specific trigger that activates the stem cell transcriptional networks and the subsequent recovery of the RAM. Among the factors that quickly respond after wound induction in the RAM are three ETHYLENE RESPONSE FACTOR (ERF) family members: *ERF109, ERF114,* and *ERF115*. They are important for the replenishment of lost cells after wound induction and have been extensively studied in this context (Bisht et al., 2023; Canher et al., 2020; Heyman et al., 2013, 2016; Hoermayer et al., 2020; Marhava et al., 2019; Zhou et al., 2019). During the regeneration, ERF115 promotes cell proliferation, cellular reprogramming, and the induction of stem cell fate (Canher et al., 2020; Heyman et al., 2013, 2016). This occurs through the activation of downstream targets PHYTOSULFOKINE PRECURSOR (PSK) *2* and *5*, WOUND INDUCED DEDIFFERENTIATION1 (WIND1), and *MP/ARF5*, respectively (Canher et al., 2020; Heyman et al., 2013, 2016). In a recent study, ERF114 and ERF115 were found to interact with three members from the GRAS-domain protein family: SCL5, SCL21, and PAT1 (Bisht et al., 2023). The combined activity of these GRAS-domain transcription factors was shown to be required for regeneration following wounding. In the same study, a DOF-type transcription factor, DOF3.4, was identified as a downstream regulator of these GRAS-domain transcription factors

and, in turn, controls periclinal cell division through the activation of *CYCD3;3* (Bisht et al., 2023).

Auxin and the plant defence hormone jasmonate (JA) have a synergistic effect on the activation of the *ERF115* transcription factor. The stress hormone JA is produced within minutes after wounding and activates the expression of ERF109, which in turn stimulates the expression of both *ERF115* and *CYCD6;1* (Zhou et al., 2019).

Wounding the root generally disturbs the auxin flux and, as such, affects the accumulation of auxin. Expression of *ERF115* is regulated by auxin after wounding but is not the primary trigger of its expression (Hoermayer et al., 2020). Auxin signalling also coordinates the wound responses by regulating cell division rates, cell expansion rates, and wound signal transduction through activation of the *ERF115* transcription factor (Hoermayer et al., 2020).

### 5.1. The stem cell masters: Auxin-PLT-SHR-SCR and maybe others. Are they the signals?

The two first 'classics' established the need for a signal from the QC for stem cell maintenance. Follow-up studies have highlighted the important role of transcription factors in this process. The transcription factors PLETHORA (PLT), and the GRAS families SHORT-ROOT (SHR)-SCARECROW (SCR) are important regulators of the stem cell niche (Figure 4) (Aida et al., 2004; Di Laurenzio et al., 1996; Galinha et al., 2007; Helariutta et al., 2000; Sabatini et al., 2003; Sozzani et al., 2010). The repatterning process leading to a new organiser after laser ablation depends on PLTs, SHR, and SCR (Xu et al., 2006). With auxin having an important role in tissue repatterning (Sabatini et al., 1999), the activity of PLTs was proposed to depend on the Auxin Responsive Factors (ARF) (Aida et al., 2004). It has also been shown that the PLTs have a gradient distribution and act in a dose-dependent manner to control different functions along the meristem (Galinha et al., 2007). Their movement was shown to be important in maintaining root zonation together with auxin (Mähönen et al., 2014). Furthermore, the transcription of *PLT* is mainly confined to the stem cell niche by PLT-induced activation of *MIR396,* which in turn induces *GROWTH-REGULATING-FACTOR (GRF)* that acts as a repressor of PLT in transit-amplifying cells (Figure 1) (Rodriguez et al., 2015). PLT protein stability is also dependent on GLV ROOT GROWTH FACTOR (RGF)/GOLVEN (GLV) peptide signalling (Matsuzaki et al., 2010; Zhou et al., 2010).

The work of Sabatini et al. discussed above illustrates the importance of auxin in the formation and maintenance of the RAM. Over the last decades, studies on auxin regulation of the RAM have taken up a prominent position in plant science. The extent to which auxin transporters control the auxin flux within the root and define the auxin maxima has been of major interest. The asymmetrically disturbed PIN proteins are essential in coordinating the auxin flux. Single mutants of PIN proteins only show mild defects, whereas higher-order mutants severely affect root patterning (Blilou et al., 2005). Modelling of auxin fluxes based on the PIN localisation revealed that the PIN-mediated auxin flux is responsible for defining the auxin gradient and robust localisation of the auxin maxima (Grieneisen et al., 2007). This proposed auxin gradient was later experimentally confirmed using tagged cell type–specific lines combined with mass spectrometry, which allowed quantification of the auxin distribution in the root (Petersson et al., 2009).

Follow-up studies on auxin transport showed that the ABC transporters also mediate polar auxin transport (Geisler et al., 2005; Geisler & Murphy, 2006; Kamimoto et al., 2012; Zhang et al., 2018).

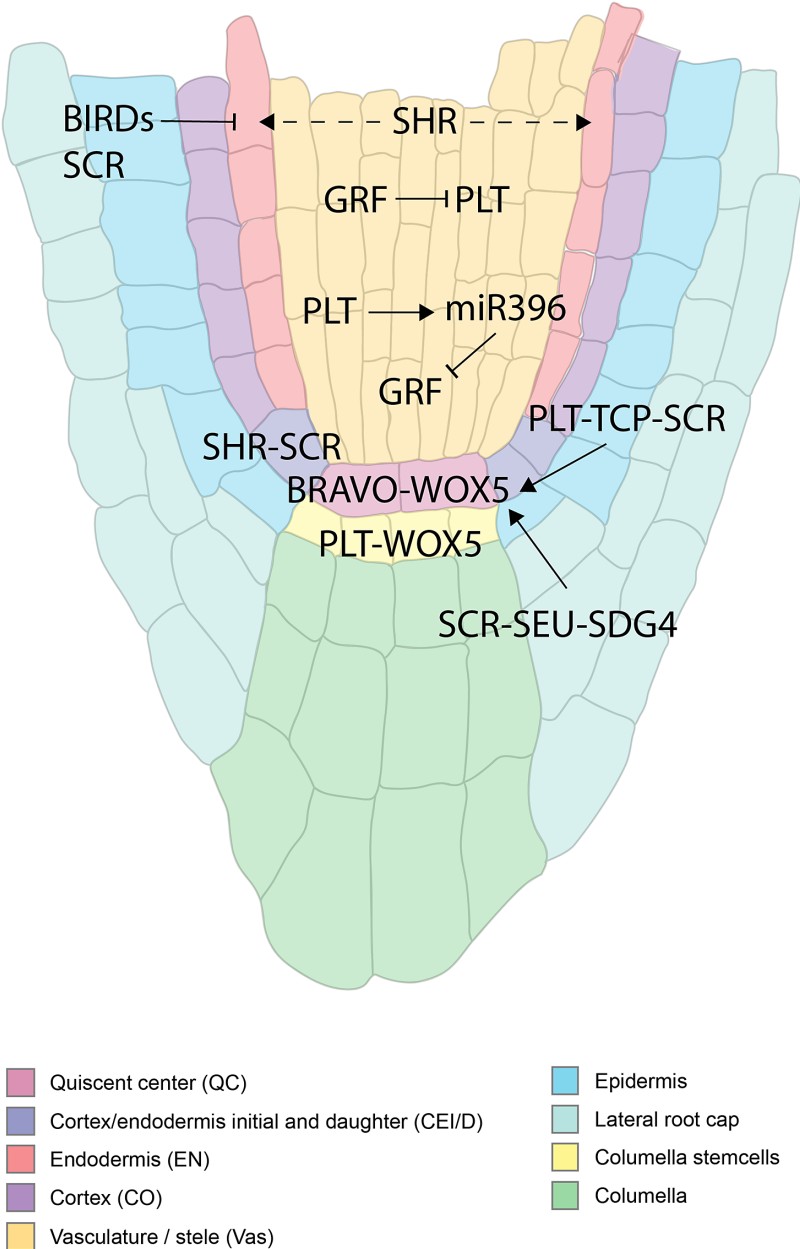

**Figure 4.** A spatial overview of the molecular players within the RAM involved in patterning and stem cell specification. Solid black arrows indicate transcriptional regulation. Dashed arrows indicate protein movement.

The PIN efflux carriers readjust the auxin flux only after wounding and not directly in response to changes in auxin distribution. Besides auxin fluxes in the root, local biosynthesis of auxin within the QC has also recently been shown to be important to maintain indeterminate growth of the root (Brumos et al., 2018).

## 6. Mobile SHR and WOX5

Signalling from the QC also involves protein movement; for instance, SHR is produced in the stele and moves one cell layer outwards through plasmodesmata to define the QC, stem cells, and the ground tissue cell layers (Gallagher et al., 2004; Nakajima et al., 2001; Vatén et al., 2011). Interestingly, the SHR signalling is controlled by SCR and BIRD family members of transcription factors through nuclear sequestration (Cui et al., 2007; Gallagher

et al., 2004; Long et al., 2015, 2017). Interaction between these members, auxin, and the RETINOBLASTOMA-RELATED protein (RBR) confines the expression of *CYCD6;1* within the cortex endodermis initial cell (Cruz-Ramírez et al., 2012). This mechanism restricted the formative divisions within the ground tissue (Cruz-Ramírez et al., 2012). Establishing a protein interaction map showed that protein complexes deploy cell type–dependent conformational changes within the stem cell niche and in mature tissues (Long et al., 2017). These complexes differentially induce gene expression, leading to the specification of distinct cell fates (Long et al., 2017). Modelling of the SHR-SCR protein complex combined with fluorescence spectroscopy technologies showed that high levels of SHR-SCR promote divisions of the CEI and repress divisions of the QC (Clark et al., 2020). This is another example of how signalling in the QC controls cell fate specification.

Another signalling pathway that has been described involves the Class I members of the teosinte-branched cycloidea PCNA (TCP) transcription factors family. PLT-TCP-SCR complexes regulate the expression of *the WUSCHEL-LIKE HOMEOBOX5 (WOX5)* gene to maintain the stem cell niche (Figure 4) (Shimotohno et al., 2018). Loss of *WOX5* results in the differentiation of the columella stem cells and is marked by the accumulation of starch granules. Additionally, the loss of WOX5 leads to an increased division of the QC, which can be suppressed in mutants of *cycd1;1* or *cycd3;3*, which shows their contribution to maintaining the quiescence of the QC (Forzani et al., 2014). *WOX5* is highly expressed in the QC, and the protein moves towards the columella stem cells (Berckmans et al., 2020; Pi et al., 2015; Sarkar et al., 2007). Expression of *WOX5* is restricted to the QC by CLE40/ACR4, ARF10, ARF16, and ROW1 (Ding & Friml, 2010; Stahl et al., 2009, 2013; Zhang et al., 2015). *WOX5* expression was also shown to be regulated by the transcriptional complex comprising SCR, the glutamine (Q)-rich SEUSS protein, and the ASH1-RELATED 3 (ASHR3) methyltransferase SET DOMAIN GROUP 4 (SDG4) (Zhai et al., 2020). SEUSS was shown to interact with SCR, and once the complex binds to the WOX5 promoter, SEUSS then recruits SDG4 to induce trimethylation of histone H3 lysine (K) 4 (H3K4me3) and subsequently activate *WOX*5 expression (Zhai et al., 2020).

As discussed in van den Berg et al. (1997), some signal from the QC is non-cell autonomously controlling differentiation of the columella stem cells (CSC). Blocking symplastic transport in the QC results in starch accumulation in the QC and CSC, leading to altered cell fate cells surrounding the QC. Furthermore, it affects the positioning of the local auxin maxima and PLT peak expression (Liu et al., 2017). Communication between QC and its neighbouring cells is crucial to maintain the SCN. It was proposed that WOX5 could act as a moving signal from the QC to inhibit differentiation (Pi et al., 2015; Sarkar et al., 2007). However, a recent publication has challenged this view (Berckmans et al., 2020). This study showed that WOX5 movement is not required for CSCs maintenance and proposed that factors independent of WOX5 might control this process.

Recent work has shown that besides transcriptional regulation, these transcription factors also form protein complexes that localise to nuclear bodies in the columella stem cells, which could serve an important role in CSC fate determination (Burkart et al., 2022). Besides PLT, *WOX5* expression is also dependent on the BRASSI-NOSTEROIDS AT VASCULAR AND ORGANIZING CENTER R2R3- MYB transcription factor (BRAVO), which has been suggested to act through the formation of a hetero dimer complex and through subsequent disruption of the WOX5 negative feedback on itself (Betegón-Putze et al., 2021; Mercadal et al., 2022). This, in turn, activates *BRAVO* expression, and as such, this mechanism confines the expression of these transcription factors to a small domain in the stem cell niche.

## 7. Perspectives

The root development community has grown tremendously in the past two decades, and our understanding of the root apical meristem has increased. Many molecular players involved in RAM maintenance and patterning have been identified, and our knowledge of how the genetic networks are regulated is ever-increasing. In recent years, the input of abiotic and biotic cues that fine-tune growth and development according to the environment of the developing root has seen a rise in interest. Additionally, omics approaches, and validation of these techniques are becoming more potent, starting with the global transcriptomic that lead to the first Arabidopsis root atlas expression map (Birnbaum et al., 2003; Nawy et al., 2005; Brady et al., 2007; Li et al., 2016) and currently the single-cell technologies in Arabidopsis and other species (Efroni et al., 2016; Guillotin et al., 2023; Nolan et al., 2023; Shahan et al., 2021; Zhang et al., 2019).

This, combined with the in vivo studies of protein–protein interactions and the emerging spatial omic technologies, will certainly contribute to a better understanding of how protein complexes and metabolites contribute to cell fate specification in the RAM. The technological advances over the past decades have allowed for quantifying cell shape, gene expression, and morphogen concentrations in space and time at an ever-increasing resolution. Combining this information with the relative position of cells within the organ using computational tools, including deep learning methods, to 3D segmentation of developing organs will allow for a better understanding of how positional information controls patterning.

**Authorship contribution.** W.S. and I.B. wrote the article.

**Financial support.** This work was supported by the KAUST Baseline Research Funds (BAS/1/1081-01-01) and by the KAUST Global Fellowship Program under the auspice of the Vice President for Research.

**Competing interest.** We have no conflicts of interest to disclose.

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
