## [Reviewer Report]

Dear Olivier,

Please find in the upload our review draft entitled “ A blast from the past: understanding stem cell specification in plant roots using laser ablation” by Smet and Blilou. This review is written upon your invitation, and it discusses three papers that, in our opinion, have contributed to major understanding in stem cell specification and maintenance in the Arabidopsis root meristems. We reviewed the following publications: Cell fate in the Arabidopsis root meristem is determined by directional signaling (van den Berg et al., 1995); Short-range control of cell differentiation in the Arabidopsis root meristem (van den Berg et al., 1997); An Auxin-Dependent Distal Organizer of Pattern and Polarity in the Arabidopsis Root (Sabatini et al., 1999).

We describe how simple experimental designs can pave the way toward a major understanding of a biological process.

We greatly appreciate your time and consideration and look forward to your response.

Sincerely Yours,

Ikram Blilou,

---

## [Reviewer Report]

The editor has carefully considered the reviewer’s comments and agrees with both that the scientist’s contributions to stem cell and cell fate specification in plants are significant. However, they suggest that a different format and a more focused approach on a broader audience could enhance the review’s impact. The reviewers also recommend integrating recent advances in the field and providing new perspectives to increase the review’s novelty, while acknowledging the continued impact of the scientist’s work. Furthermore, the reviewers note minor issues with the figures, which could be improved by explicitly showing each paper’s contribution to the depicted knowledge, and suggest weaving short paragraphs into more descriptive ones for greater impact. The organization could also be clearer in distinguishing between subsections of the Scheres work and a more encompassing literature review. Lastly, grammar issues should be addressed.

---

## [Reviewer Report]

Dear Editors,

Please find in the upload the revised version of the classic review entitled “ A blast from the past: understanding stem cell specification in plant roots using laser ablation”.

We thank the editors and the reviewers for their constructive comments that have helped shape the review and improve its quality.

We have implemented the following changes.

1- We have included figures highlighting the key findings from each classical paper

2- We are also including a response to each point raised by the reviewers.

We thank you for this opportunity and look forward to your response

---

## [Reviewer Report]

Dear Olivier,

Thank you for the acceptance of our review.

Please find in the upload the second revision of the review with all the comments and reviewer’s suggestions implemented.

With best wishes,

Ikram

---

## [Reviewer Report]

Thank you for your review, and thank you for your understanding for any delay during the review process.

Best,

Ross Sozzani